# Identification of CTLA-4-Positive Cells in the Human Tonsil

**DOI:** 10.3390/cells10051027

**Published:** 2021-04-27

**Authors:** Markus Tiemann, Dmitri Atiakshin, Vera Samoilova, Igor Buchwalow

**Affiliations:** 1Institute for Hematopathology, Fangdieckstr. 75a, 22547 Hamburg, Germany; mtiemann@hp-hamburg.de (M.T.); verasamoilova@hotmail.com (V.S.); 2Research and Educational Resource Center for Immunophenotyping, Digital Spatial Profiling and Ultrastructural Analysis Innovative Technologies, Peoples’ Friendship University of Russia, 117198 Moscow, Russia; earth-mars38@yandex.ru; 3Research Institute of Experimental Biology and Medicine, Burdenko Voronezh State Medical University, 394036 Voronezh, Russia

**Keywords:** CTLA-4, antibodies, tonsil, T-lymphocytes, T-cell activation, immunophenotyping

## Abstract

CTLA-4 (cytotoxic T-lymphocyte-associated protein 4) was originally defined as a T-lymphocyte antigen and was used as a target in cancer immunotherapy. Unfortunately, the existence of CTLA-4 in cells other than T-lymphocytes is often overlooked. The goal of the present study was to analyze the distribution pattern of CTLA-4 in the human tonsils using a panel of anti–CTLA-4 antibodies of different clones. We found that CTLA-4 was expressed in T-lymphocyte cells of various geneses, including hematopoietic cells and their derivatives (monocytes, macrophages, dendritic, plasma cells, mast cells, and neutrophils), as well as stromal cells of mesodermal (mesenchymal) origin and reticular epithelial cells of ectodermal origin. The expression of CTLA-4 in cells of different origins supports the proposition that CTLA-4 is not restricted to the lymphoid cell lineage and can provide broader effects of CTLA-4 on immune regulation.

## 1. Introduction 

CTLA-4 (cytotoxic T-lymphocyte-associated protein 4) is a protein receptor that functions as an immune checkpoint. Brunet et al. (1987) were the first to identify CTLA-4 [1] and reported that it is constitutively expressed in regulatory T-cells and upregulated in conventional T-cells after activation in inducible models [1]. Thus, CTLA-4 was originally defined as a T-lymphocyte antigen [1] but was later described as a T-cell surface receptor [2,3,4,5]. 

In 1994, it was discovered that CTLA-4 can function as a negative regulator of T cell activation [6]. Following this, Allison’s research group [7,8] found that CTLA-4 acts similarly to PD-1 [9] as an inhibitory molecule to restrict T cell responses. When bound to its ligands CD80 or CD86 on the surface of various immune cells including B cells, monocytes and antigen-presenting cells, such as macrophages and dendritic cells, CTLA-4 downregulates immune responses preventing the immune system from killing cancer cells. Allison [7] discovered that the blockade of CTLA-4 with monoclonal antibodies restricted the binding of CTLA-4 to its ligands and increased anti-tumor immune responses with tumor rejection [7,10,11].

The blockade of the checkpoint molecule CTLA-4 with monoclonal antibodies based on Allison’s research [7,8] enabled the development of breakthrough therapies in oncology and ultimately led to the clinical development of Ipilimumab (trade name Yervoy) [12]. The use of Ipilimumab for melanoma treatment was based on Allison’s concept that Ipilimumab binds to CTLA-4 on the surface of T cells and blocks the inhibitory signal, thereby releasing the effector function of the T-lymphocytes [13].

In contrast to this early paradigm, other studies over the last two decades have demonstrated both constitutive and inducible expression of CTLA-4 in a broad distribution of tissues and cell types other than lymphocytes, particularly dendritic cells [14], human breast tumor cells [15], and macrophages [16]. Of special interest is the paper by Pistillo et al. [17], who first described the reactivity and function of CTLA-4 in different cell types. These and other publications are discussed in the review by Oyewole-Said et al., [18]. What these works have in common is that they were performed on isolated cells using FACS, WB, RT-PCR, and flow cytometry, while none were performed with the use of immunohistochemical staining on tissue sections. Immunohistochemical approaches were applied only in single studies reviewed by Oyewole-Said et al. [18], and in some of these reports anti-CTLA-4 antibodies of clones BNI3 and 14D3 were used. However, the antibody manufacturer Abcam removed its recommendation to use the anti-CTLA-4 antibody (clone BNI3, #ab19792) in immunohistochemical approaches following customer feedback (https://www.abcam.com/ctla4-antibody-bni3-ab19792.html, accessed on 15 April 2021). Likewise, another clone of the anti-CTLA-4 antibody (clone 14D3) that was used in studies reviewed by Oyewole-Said et al. [18] is recommended for FC/FACS, but not for immunohistochemical applications (https://www.citeab.com/antibodies/2039829-16-1529-cd152-ctla-4-monoclonal-antibody-14d3-f, accessed on 15 April 2021). In contrast to the studies mentioned above, the aim of our study was to analyze the distribution pattern of CTLA-4 in cells other than T-lymphocytes in tissue sections in situ.

As it is known that the interaction between CTLA-4 and its ligands restricting T cell responses occurs in lymphoid organs [19,20], our analysis was carried out on human tonsils. We performed immunophenotyping of CTLA-4-positive cells using multiple fluorescent immunolabeling of this protein vs. a panel of antibodies raised against the cells of various geneses, including myelopoiesis cells and their derivatives (monocytes, macrophages, dendritic, plasma cells, mast cells, and neutrophils), stromal cells of mesodermal (mesenchymal) origin, and reticular epithelial cells of ectodermal origin. Our findings that CTLA-4 is expressed in cells of different origins supports the proposition that CTLA-4 is not restricted to the lymphoid cell lineage and can provide broader effects of CTLA-4 on immune regulation.

## 2. Materials and Methods

### 2.1. Case Selection

Tissue samples were removed for diagnostic purposes. Tonsil tissue was obtained from four male and six female patients undergoing tonsillectomy for recurrent tonsillitis. The age range of the patients was 8–27 years. Samples were retrieved from the files of the Institute for Hematopathology, Hamburg, Germany. The staining of cell subsets that can express CTLA-4 among ten different donors was examined by three pathologists (MT, IB and DA) independently. Upon reaching a consensus, no significant differences between the probes under study both in cytomorphology and immunophenotyping were found.

The samples were redundant clinical specimens that had been anonymized. This study was conducted in accordance with the principles of the World Medical Association Declaration of Helsinki “Ethical Principles for Medical Research Involving Human Subjects” and was approved by the Institutional Review Board of the Institute for Hematopathology, Hamburg, Germany.

### 2.2. Tissue Probe Stainings

Tissue probes left over during the routine diagnostic procedure were fixed in buffered 4% formaldehyde and routinely embedded in paraffin. Paraffin tissue sections (2 µm thick) were deparaffinized with xylene and rehydrated with graded ethanols according to a standard procedure [21].

### 2.3. Immunohistochemistry

For the immunohistochemical assay, deparaffinized sections were subjected to antigen retrieval by heating the sections in a steamer with R-UNIVERSAL Epitope Recovery Buffer (Aptum Biologics Ltd., Southampton, UK), at 95 °C for 30 min. Blocking the endogenous Fc receptors prior to incubation with primary antibodies was omitted according to our earlier recommendations [22]. After antigen retrieval and, when required, quenching endogenous peroxidase, sections were immunoreacted with primary antibodies. The list of primary antibodies used in this study is presented in Table 1. Immunohistochemical visualization of bound primary antibodies was performed either with Ventana Slide Stainer or manually according to the standard protocol [21,23]. For manually performed immunostaining, primary antibodies were applied in concentration from 1 to 5 µg/mL and incubated overnight at +4 °C. For visualization of primary anti-CTLA-4 antibodies, we used tyramide signal amplification (TSA) [21].

Bound primary antibodies were visualized using secondary antibodies (purchased from Dianova, Hamburg, Germany, and Molecular Probes, Darmstadt, Germany) conjugated with Cy3, Alexa Fluor-488, or Cy5. The final concentration of secondary antibodies was between 5 and 10 µg/mL PBS. Single and multiple immunofluorescence labelling were performed according to standard protocols [21]. The list of secondary antibodies and other reagents used in this study is presented in Table 2.

To simultaneously detect antigens from the same host species, we performed TSA with a subsequent heat elution treatment after each immunostaining step [24]. The bound primary/secondary antibody complex from the preceding immunolabeling step was eluted with a citrate/acetate-based buffer at pH 6.0, containing 0.3% SDS (also available from VENTANA as CC2 solution, cat # 950-223) [25]. Nuclei were counterstained with 4′,6-diamidino-2-phenylindole (DAPI, 5 µg/mL in PBS) for 15 s, and the sections were then mounted using VectaShield (Vector Laboratories, Burlingame, CA, USA).

### 2.4. Quantification of CTLA-4-Positive Cells in Tonsillar Compartments

Five to ten sections were prepared from each tissue block, and the numbers of CTLA-4(+) cells in each ware determined by counting the cells within at least 50 fields of view (350 × 250 µm^2^) using a ×40 objective lens on each section. The quantification of the cell content was performed using a counting program incorporated in the AxioVision software (Carl Zeiss Vision, Oberkochen, Germany). 

The intensity of staining in tonsillar compartments (Table 3) was reviewed by three pathologists independently (MT, IB, and DA) using a ×20 objective lens and, upon reaching consensus, the scoring was assigned from (+++) to (−). A consensus scored of (+++) represented the largest number of cells with CTLA4 expression in the field of view with the number of CTLA-4-positive cells being more than 50. When the number of cells averaged between 20 and 50, the average score was assigned as (++). When 5–20 cells were detected, a low content of cells was set as (+). In the cases of detecting five cells or fewer, the “rare single cells” score was assigned as (−).

### 2.5. Controls

Control incubations were omission of primary antibodies or substitution of primary antibodies by the same IgG species (Dianova, Hamburg, Germany) at the same final concentration as the primary antibodies. The exclusion of either the primary or the secondary antibody from the immunohistochemical reaction, or the substitution of primary antibodies with the corresponding IgG at the same final concentration resulted in a lack of immunostaining. The TSA step alone did not contribute to any specific immunostaining that might have influenced the analysis. Furthermore, the specific and selective staining of different cells with the use of primary antibodies from the same species on the same preparation can be considered a sufficient control for the immunostaining specificity.

### 2.6. Image Acquisition

Immunostained tissue sections were observed on a Zeiss Axio Imager. Z1 configured for fluorescence microscopy and equipped with a microscope camera Zeiss Axio Cam HRm, monochrome CCD. The images were acquired at 96 DPI and submitted with the final revision of the manuscript at 300 DPI.

### 2.7. Data Availability

The authors declare that the data supporting the findings of this work are available within the article or from the corresponding author upon reasonable request.

## 3. Results

Since most of the available data on CTLA-4 expression are limited to human T cells, the goal of the present study was to investigate CTLA-4 localization in cells of various origins using a panel of anti–CTLA-4 antibodies of different clones targeting different CTLA-4 epitopes. 

Here, we analyzed the distribution pattern of CTLA-4 in the human tonsil using anti–CTLA-4 antibodies of four different clones (UMAB249, AA 57-86, SP355, and CAL49). The number of CTLA-4(+) cells and the distribution pattern of CTLA-4 in tonsillar compartments varied strongly with anti-CTLA-4 antibodies of different clones (Table 3), and immunophenotyping of CTLA-4(+) cells revealed the expression of CTLA-4 in cells of various origin. 

### 3.1. Immunomorphological and Histotopographic Characteristics of CTLA-4-Positive Cells Identified with Antibodies of the SP355 Clone

CTLA-4(+) cells visualized with the anti–CTLA-4 antibody of SP355 clone were detected mainly in the extrafollicular regions of the tonsil, including the connective tissue of the organ (Table 3, Figure 1a). They were localized in the lymphatic follicles, including germination centers (Table 3, Figure 1b), less often. Considering the intensity of immunopositivity, these cells could be divided into two groups: either high or moderate CTLA-4 content (Figure 1c). CTLA-4(+) cells were mostly located separately, but in some cases, they were in contact with each other (Figure 1d,e). Regardless of the CTLA-4 level of expression assayed with the antibody of this clone, CTLA-4(+) cells lacked a co-expression with cytomarkers of CD3 and CD8 T-lymphocytes, even though a co-expression with CD4 was in rare cases detected (Figure 1f–h).

The cells in which CTLA-4 expression was most pronounced were usually dispersed in the germinative centers and mantle zone of lymphatic follicles, as well as in extrafollicular lymphoid tissue (Figure 1a,b). CTLA-4(+) material filled the cytoplasm diffusely with intense uniform staining; however, certain cell compartments were sometimes selectively contoured. The nucleus lay eccentric, with a characteristic arrangement of heterochromatin lumps (Figure 1i and Figure 2a,b). These CTLA-4(+) cells showed a high immunopositivity for CD38 (Figure 1i), which was less pronounced for CD45 (Figure 2a), and in some cases of CD138 (Figure 2b). Judging by the available differentiation clusters and morphological data, these cells can be characterized as plasma cells.

Cells with moderate immunopositivity for CTLA-4 were more numerous than cells with a high level of CTLA-4 expression. Most were detected in the region of connective tissue septa, although to a lesser extent they were found in extrafollicular cords of lymphoid tissue, in the subepithelial region of the tonsillar mucosa plate, and directly in the reticular epithelium (Figure 1a,b and Figure 2c,d). These cells varied in size, but their main feature was the formation of granule-like intracytoplasmic formations of CTLA-4(+) material, together with diffuse staining of the cytoplasm (Figure 2c,d). The nucleus was often located centrally in the cell. Sometimes exocytosis patterns of microvesicular cytoplasmic formations in the extracellular matrix were revealed. In such cells, a highly manifested CTLA-4 co-localization was detected with tryptase, which was detected mainly in granules (Figure 2c,d). Thus, immunomorphological and histological criteria enabled these CTLA-4(+) cells to be classified as mast cells. Most CTLA-4(+) cells contained CD117, and some of them co-expressed CD45. Different degrees of expression of CTLA-4(+) material in mast cells, as well as a significant content of CTLA 4+ cells within the mast cell population, suggest their variable role in the regulation of immune responses. Thus, characterization of the cell pool in the tonsil with which CTLA-4(+) cells come into contact may be of great importance. In particular, the CTLA-4(+) cells were characterized by contacting lymphocytes (Figure 1f–h and Figure 2e), plasmacytes (Figure 2f), macrophages (Figure 2g–i), and NK cells (Figure 2j), which emphasizes their involvement in immune responses. The contacts between CTLA-4(+) cells and undifferentiated hematopoietic cells (Figure 2k) and elements of the microvasculature (Figure 2l) are also important.

### 3.2. Immunomorphological and Histotopographic Characteristics of CTLA-4-Positive Cells Identified with Antibodies of the UMAB249 Clone

Cells immunostained with antibodies to the CTLA-4 (UMAB249 clone) were predominantly detected in the extrafollicular regions of the tonsil epithelium, both in the reticular epithelium and in the integumentary stratified squamous non-keratinized epithelium (Table 3, Figure 3 and Figure 4). These cells were small to medium in size (from 8 to 12 µm), round or oval, with a layer of immunopositive cytoplasm surrounding the nucleus (Figure 3a′,c,d and Figure 4a–d). However, some cells were of irregular form or elongated and could form outgrowth-like formations (Figure 3c,d and Figure 4e,f). In the interfollicular lymphoid tissue, these cells sometimes formed clusters in which they could come into contact with each other (Figure 4d). In the tonsil integumentary epithelium, CTLA-4(+) cells could be located both in the basal region and in the surface layers (Figure 3b,e). In some cases, CTLA-4-immunopositive areas of the epithelium were revealed that extended to the entire epithelium thickness.

The immunophenotype of CTLA-4(+) cells for this clone was characterized by a lack of expression of the common leucocyte antigen CD45, as well as of CD3, CD4, and CD8 (Figure 3a and Figure 4a). At the same time, almost all CTLA-4(+) cells located in the stroma of the tonsils revealed a vimentin expression (Figure 3e). Interestingly, in the CTLA-4(+) reticular epithelium, the cells could be divided into two groups: those expressing vimentin or cytokeratin (Figure 3b–e). Furthermore, CTLA-4(+) cells were found to co-express podoplanin (Figure 3e and Figure 4e), which was most common in the epithelium. In the interfollicular region, CTLA-4(+) cells sometimes showed the expression of CD1a and CD14 and, less often, CD68 and CD16 (Figure 4d,f,g). Such an immunophenotype implies that these CTLA-4(+) cells may belong to the stromal cellular component of mesodermal origin, or to the ectoderm derivatives located exclusively in tonsillar reticular epithelial cells. The presence of antigen-presenting properties, as well as NK cell markers in some of the CTLA-4(+) cells, must also be considered. From the point of view of the histotopography of these cells in a specific tissue microenvironment, their frequent contact with CD3 and CD4 positive lymphocytes, as well as with antigen-presenting cells, in particular CD1a, CD14, CD163 (Figure 4c,f), and B-lymphocytes, implies their active participation in the development of immune responses.

### 3.3. Immunomorphological and Histotopographic Characteristics of CTLA-4-Positive Cells Identified with Antibodies of the CAL49 Clone

The population of CTLA-4(+) cells detected with the antibody of the CAL49 clone was the most numerous (Table 3) and had the smallest size compared to the CTLA-4(+) cells detected by antibodies of other CTLA-4(+) clones. They were predominantly CD3- and CD4-positive (Figure 5) and were located mainly within the lymphatic follicles and extrafollicular lymphoid tissue (Figure 5a), but they could also infiltrate the reticular epithelium (Figure 6a,b). Similar to plasma cells, a close co-localization of CTLA-4(+) cells with podoplanin-immunopositive structures was observed in the lymphatic follicles (Figure 6c). As expected, the vast majority of these cells contained a CD45 differentiation cluster. CTLA-4(+) material in the cell cytoplasm was located in the form of a narrow, unevenly colored perinuclear rim; CTLA-4(+) material was presented in the cytoplasm in the form of granule-like formations (Figure 5d). It is noteworthy that most CTLA-4(+) cells co-expressed CD4 and only some of the CD8 (Figure 5e–g and Figure 6d–f). Some of the CTLA-4(+) cells did not express CD3 (Figure 5d), which suggests the expression of CTLA-4 detected with antibodies of this clone in some other unidentified cells in the tonsil.

Close contact between CTLA-4(+) cells was observed with CD14(+) cells within the epithelium (Figure 6b,g). An unusual feature of these contacts was the frequent formation of certain “niches” for CTLA-4(+) lymphocytes, in which they found themselves surrounded by the cytolemma of antigen-presenting cells (Figure 6b–g). In extrafollicular cords of lymphoid tissue, frequent contact with CD68(+) cells was revealed (Figure 6h). Within the germinative zone of the lymphatic follicles, CTLA-4(+) cells were often observed in close contact with plasma cells (Figure 6i).

### 3.4. Immunomorphological and Histotopographic Characteristics of CTLA-4-Positive Cells Identified with Antibodies of the AA 57-86 Clone

Anti-CTLA-4 antibodies of the AA 57-86 clone detected the cells with the smallest number of examined probes of tonsillar tissue compared with anti-CTLA-4 antibodies of other clones (Table 3). As a rule, they were located in groups in the area of the reticular epithelium and adjacent connective tissue of the lamina propria mucosa (Figure 7a), as well as in the extrafollicular regions, including the strands of lymphoid tissue around the follicles (Figure 7b). The cells that were immunopositive for CTLA-4 of this clone did not manifest immunopositivity for CD3 (Figure 7c). As expected, similar results were also obtained with the detection of CD4 and CD8. Notably, CTLA-4(+) cells were of various shapes and sizes—from medium to large (Figure 8). It was revealed that most of these cells carried the CD14 antigen (Figure 8a,b). Together with CD14, CTLA-4(+) cells were quite often characterized by the expression of CD68 and CD163 (Figure 8c,d). Sometimes granule-like CTLA-4-immunopositive cytoplasmic formations were detected in such cells (Figure 7c–e and Figure 8a,c,e). Some of the CTLA-4(+) cells possessed the general leukocyte antigen CD45, while other more numerous cells did not (Figure 7d,e). In addition, CTLA-4 expression was detected in tryptase(+) mast cells (Figure 8e).

It appears that some of the of CTLA-4 cells that were detected with AA 57-86-clone antibodies were representatives of the hematopoietic cells with signs of monocytic phagocytes—monocytes or macrophages. The CTLA-4 cells that were immunopositive for the AA 57-86 clone may be representatives of the stromal component of mesoderm origin. However, a more precise identification in the context of this study could not be determined.

## 4. Discussion

Multiple immunofluorescent stains in this study revealed that CTLA-4 was expressed within the cytoplasm of T-lymphocytes, and other cells of various geneses. Therefore, our results indicate that CTLA-4 is not restricted to the lymphoid cell lineage and can function as a target molecule for apoptosis induction. The expression of different CTLA-4 epitopes in cells of various origins indicates that these cells may also be involved in the transduction of an immunosuppressive signal. Whether these signals would follow the same inhibitory pathway typical of activated T-cells remains to be established. 

Unfortunately, it is often ignored that CTLA-4 exists also in cells other than lymphocytes. Most of the available data on CTLA-4 expression is limited to human T cells. Accordingly, we found that immunolabeling CTLA-4 with the antibody clone CAL49 vs. T lymphocytes markers CD3, CD4, and CD8 showed that CTLA-4(+) T lymphocytes in tonsils were typically CD3(+) and CD4(+), but not CD8(+), as was earlier reported by Brown et al. [26]. However, cells that were immunostained with antibodies to the CTLA-4 (UMAB249 clone) were predominantly detected in the extrafollicular regions of the tonsil epithelium—both in the reticular epithelium and in the integumentary stratified squamous non-keratinized epithelium—but not in T-lymphocytes. A decisive answer to the question of which cells serve as targets for Ipilimumab (trade name Yervoy, developed by Bristol-Myers Squibb) in cancer immunotherapy can be obtained by immunophenotyping of CTLA-4(+) cells with an anti-CTLA-4 antibody of the same clone as a monoclonal anti-CTLA-4 therapeutic antibody Ipilimumab. Further studies on this issue are warranted. 

Our findings have allowed us to draw several conclusions. Antibodies to CTLA-4 of different clones detect this protein in cells of various origins. Generally, different immunolabeling patterns in anti-CTLA-4 antibodies of different clones can be explained by generating anti-CTLA-4 antibodies that recognize different epitopes of this protein: clone UMAB 249 can be used to detect CTLA-4 in stromal cells of a specific tissue microenvironment and in the reticular epithelium; clone SP355 selectively detects plasmocytes and mast cells; clone AA 57-86 detects both hematopoietic cells and stromal elements; and clone CAL49 predominantly detects T-lymphocytes. The expression of CTLA-4 in cells of different origins supports the alternative view that CTLA-4 is not restricted to the lymphoid cell lineage and can provide broader effects of CTLA-4 on immune regulation. Thus, expansion of the existing ideas about the role of CTLA-4 may lead to the development of new approaches to improve the quality of diagnosis and provide further insights into fundamental mechanisms underlying targeted therapy of cancer and other pathological conditions.

## Figures and Tables

**Figure 1 cells-10-01027-f001:**
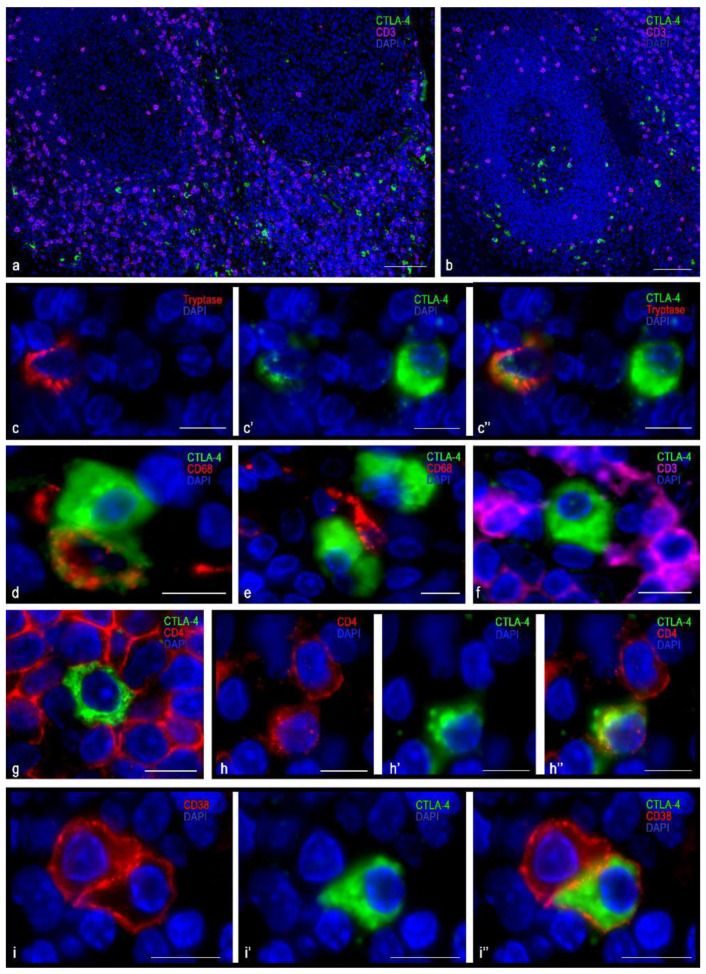
Histoarchitectonics and immunophenotype of CTLA-4(+) tonsillar cells stained with antibodies of the SP355 clone. (**a**) Predominant localization of CTLA-4(+) in the extrafollicular regions of the tonsil. (**b**) Location of CTLA-4(+) cells in the germinal center of the lymphatic follicle and in the perifollicular zone. (**c**–**c″**) Cells with moderate (tryptase) and high levels of CTLA-4 expression. (**d**) Two CTLA-4(+) cells, one of which expresses CD68. (**e**) CTLA-4(+) cells are in contact with type 1 macrophage (presumably). (**f**) CTLA-4(+) cells contacting CD3+, lack of co-expression. (**g**) CTLA-4(+) cell surrounded by CD4+ cells. (**h**–**h″**) Co-expression of CD4 and CTLA-4. (**i**–**i″**) Co-localization of CTLA-4 and CD38 (plasma cell). Scale bar 125 µm for (**a**,**b**), and 10 µm for the rest.

**Figure 2 cells-10-01027-f002:**
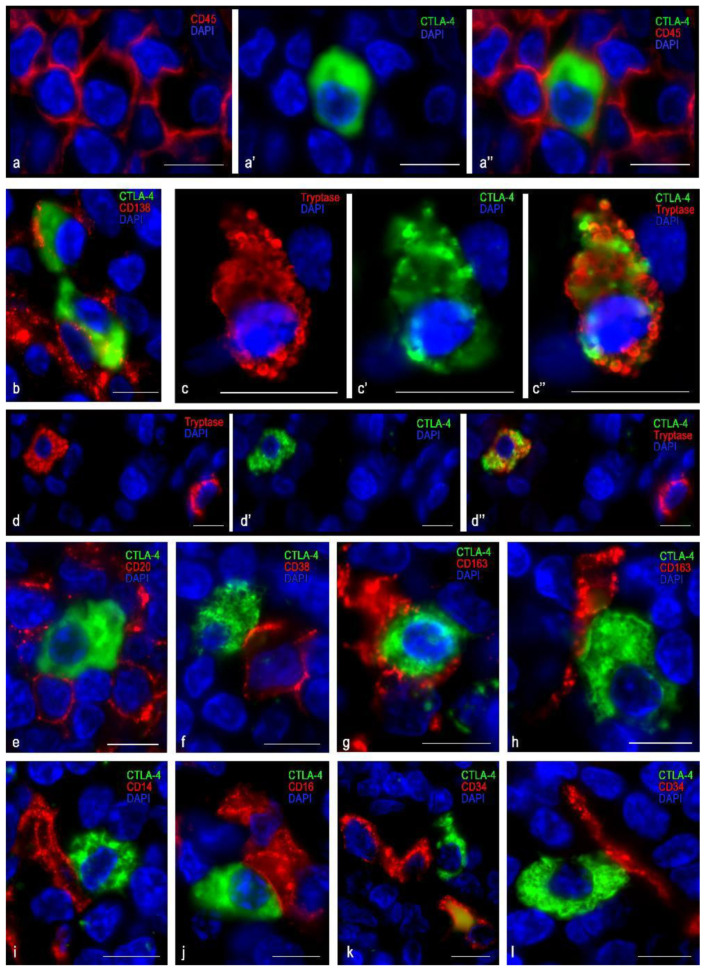
Localization of CTLA-4(+) tonsillar cells within a specific tissue microenvironment when stained with anti-CTLA-4 antibodies of the SP355 clone. (**a**–**a″**) Expression of CD45 in a CTLA-4(+) cell. (**b**) Expression of CD138 in a CTLA-4(+) cell (presumably plasma cell). (**c**–**c″**) Expression of CTLA-4 in a mast cell. (**d**–**d″**) Two mast cells, one of which expresses CTLA-4. (**e**) CTLA-4(+) cell surrounded by B-lymphocytes, but lack of expression of CD20. (**f**) CTLA-4(+) cells contacting the plasma cell (presumably). (**g**,**h**) Various options for contacting CTLA-4(+) cells with type 2 macrophages (likely). (**i**) CTLA-4(+) cells contacting a CD14(+) cell (presumably macrophage). (**j**) Close contact between CTLA-4(+) cells and an NK cell (likely). (**k**,**l**) CTLA-4(+) cells contacting a capillary and a CD34+ undifferentiated hematopoietic cell. Scale bar 10 μm.

**Figure 3 cells-10-01027-f003:**
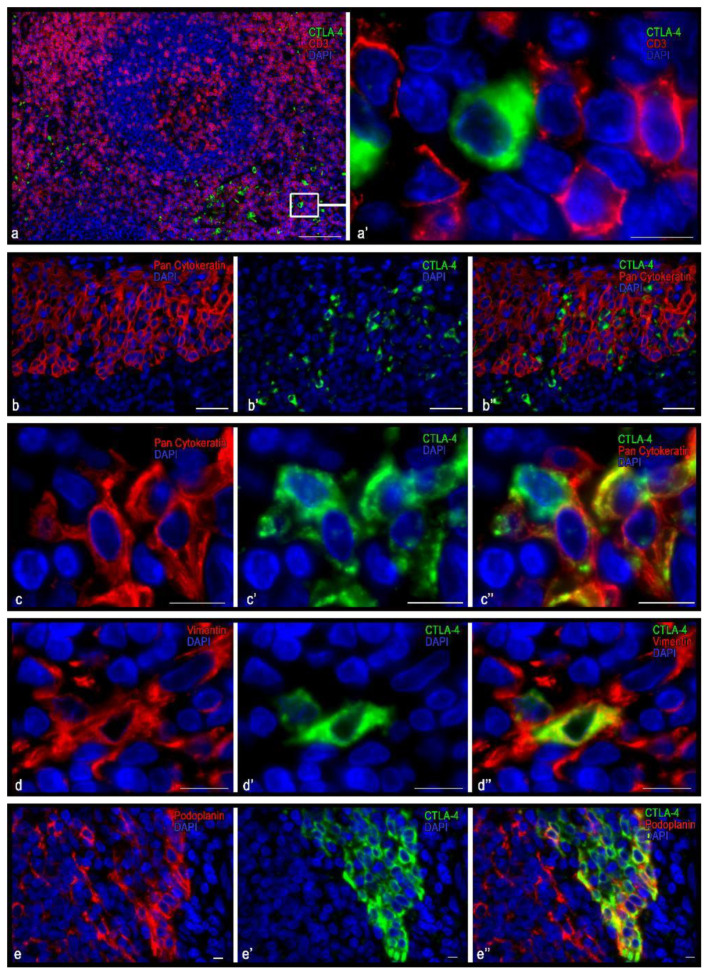
Histoarchitectonics and immunophenotype of CTLA-4(+) tonsillar cells when stained with antibodies of the UMAB249 clone. (**a**) Localization of CTLA-4(+) cells in interfollicular lymphoid cords. (**a′**) An enlarged portion of figure (**a**). CTLA-4(+) cells do not express CD3 but are adjacent to and in contact with lymphocytes. (**b**–**b″**,**c**–**c″**) CTLA-4(+) cells in the reticular epithelium at low (**b**) and high (**c**) magnification. Obviously, some CTLA-4(+) cells express cytokeratins, while others do not. (**d**–**d″**) Immunopositive cells for vimentin, two of which express CTLA-4. (**e**–**e″**) The area of the reticular epithelium, protruding into the own plate of tonsils. Epithelial cells have an intense expression of CTLA-4. Moreover, the basal layers of the epithelium, as well as some cells of the more superficial layers, have immunopositivity for podoplanin. Scale bar is 125 µm (**a**), 50 µm (**b**), and 10 µm (**a′**, and the rest).

**Figure 4 cells-10-01027-f004:**
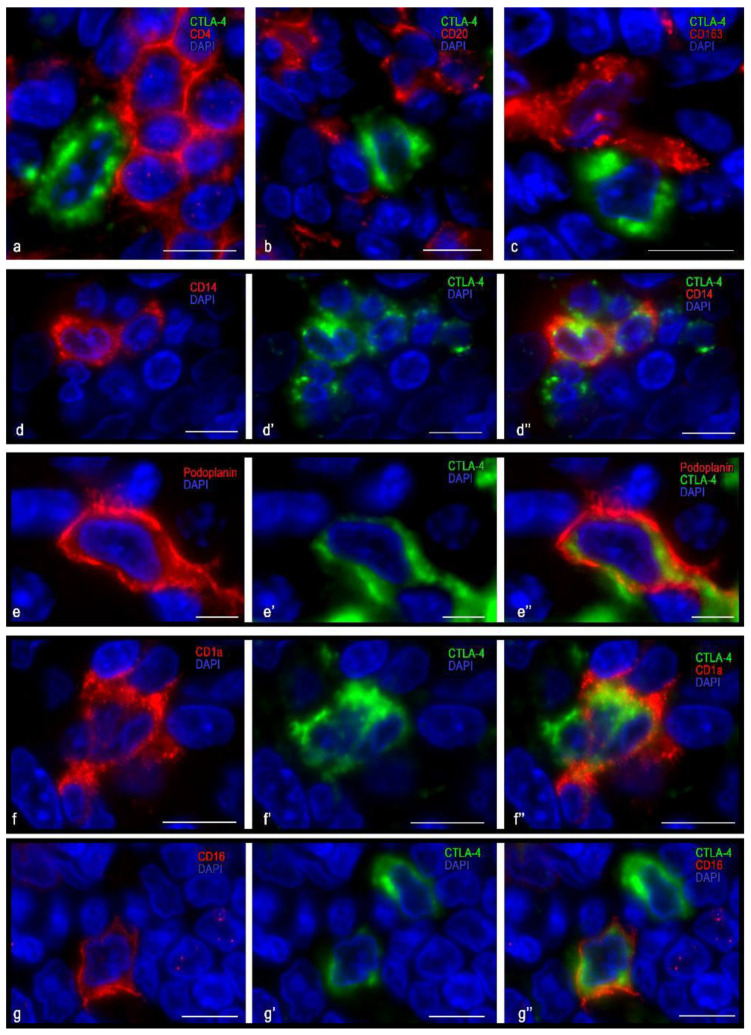
Immunomorphological and histotopographic characteristics of tonsillar cells with CTLA-4 expression stained with antibodies of the UMAB249 clone. CTLA-4(+) cells neighboring T-lymphocytes (**a**), B-lymphocytes (**b**), and macrophage of M2 type (**c**). Co-expression of CTLA-4 with CD14 (**d**–**d″**), podoplanin (**e**–**e″**), CD1a (**f**–**f″**), and CD 16 (**g**–**g″**). Scale bar: 5 µm.

**Figure 5 cells-10-01027-f005:**
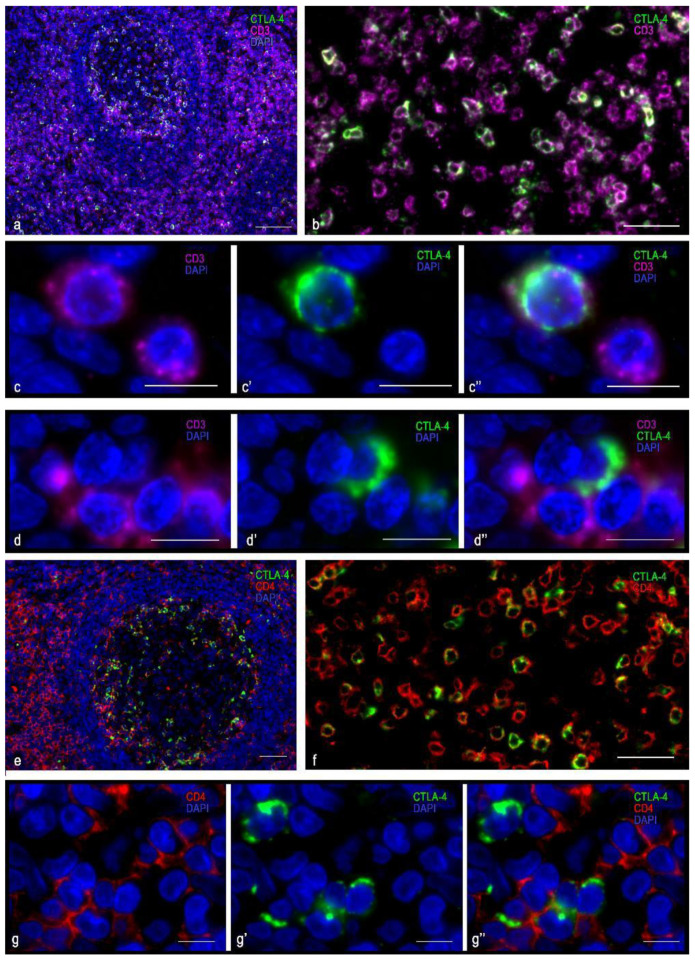
Histoarchitectonics and immunophenotype of CTLA-4(+) tonsillar cells stained with antibodies of the clone CAL49. (**a**) Histotopography of CTLA-4(+) cells in the lymphatic follicle and in the extrafollicular lymphoid tissue of the tonsil. In the lymphatic follicles, CTLA-4(+) cells are localized mainly within the germination center. (**b**) Fragment of the light zone of the germination center; the intracellular co-expression of CTLA-4 and CD3 is clearly visible. (**c**–**c″**) Cytological features of the intracellular co-localization of CTLA-4 immunopositive material and CD3 in lymphocytes. (**d**–**d″**) There was a lack of expression of CD3 in CTLA-4(+) cell. (**e**,**f**,**g**–**g″**) Co-expression variants of CTLA-4 + and CD4. The predominant location of CD4+ lymphocytes and CTLA-4(+) cells was in the germinal center of the lymphatic follicle, with most CTLA-4(+) cells expressing CD4. The presence of CD4-negative CTLA-4(+) cells (**g**) is noteworthy. Scale bar: 125 μm (**a**), 50 μm (**b**,**e**,**f**), 10 μm (the rest).

**Figure 6 cells-10-01027-f006:**
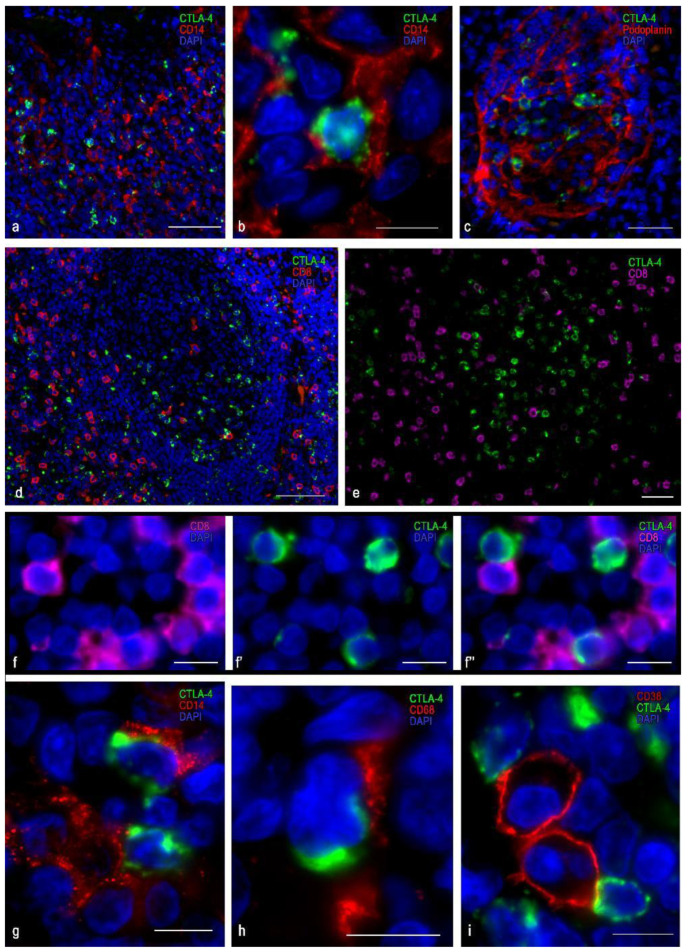
Histoarchitectonics of CTLA+ tonsillar cells when stained with antibodies of the clone CAL49. (**a**) Location of CTLA-4(+) cells in the reticular epithelium. (**b**) Formation of a “niche” for CTLA-4(+) cells. (**c**) Podoplanin-positive cells take part in the formation of a specific tissue microenvironment for CTLA-4(+) cells in the germinal center of the tonsillar lymph node. (**d**–**f**) Co-localization of CTLA-4 and CD8 in lymphocytes. The expression of CD8 in CTLA-4(+) cells is rather rare (**f**–**f″**). (**g**) CTLA-4(+) cells contacting CD14+ cells of the reticular epithelium. (**h**) Contact with a CD68(+) cell in the extrafollicular zone of the tonsil. (**i**) Co-localization of CTLA-4(+) cells and plasma cells in the germinal center of the tonsillar lymph node. Scale bar: 100 µm (a), 50 µm (**c**–**e**), and 10 µm (the rest).

**Figure 7 cells-10-01027-f007:**
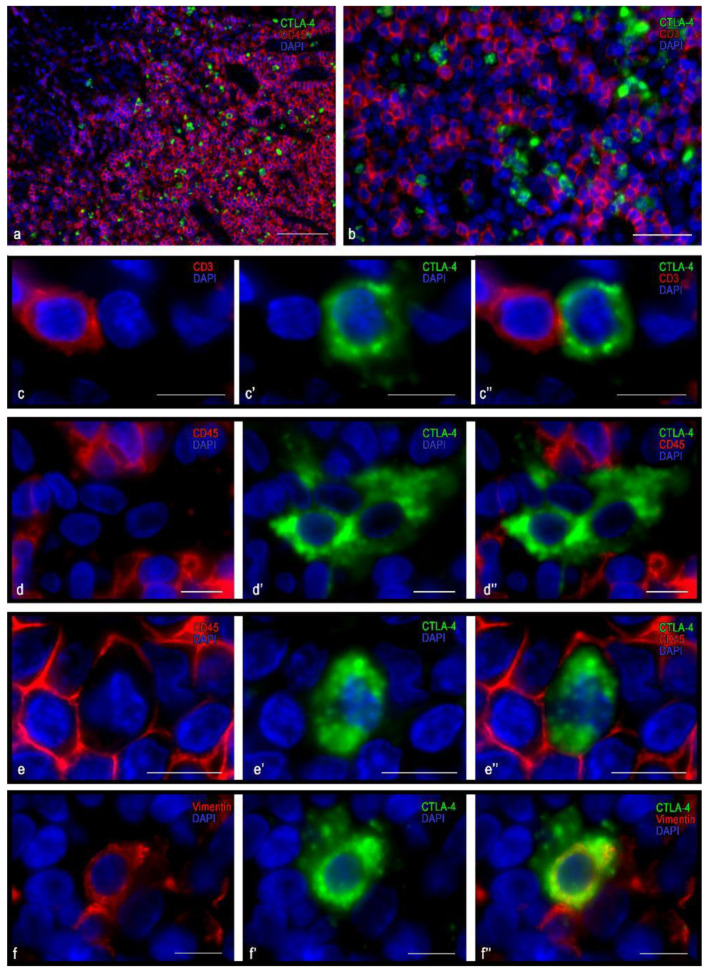
Histoarchitectonics of CTLA-4(+) tonsillar cells when stained with antibodies of the AA 57-86 clone. (**a**) Localization of CTLA-4(+) in the reticular epithelium. (**b**) Location of CTLA-4(+) cells in the interfollicular cords of lymphoid tissue. (**c**–**c″**) CD3-negative CTLA-4(+) cell in extra-follicular zone of tonsil. (**d**–**d″**) CD45-negative CTLA-4(+) cells with granulocyte-like cytoplasmic formations. (**e**–**e″**) CD45+ and CTLA-4(+) cell. (**f**–**f″**) Expression of vimentin in a CTLA-4(+) cell. Scale bar: 50 µm (**a**,**b**), 10 µm (**c**–**f**).

**Figure 8 cells-10-01027-f008:**
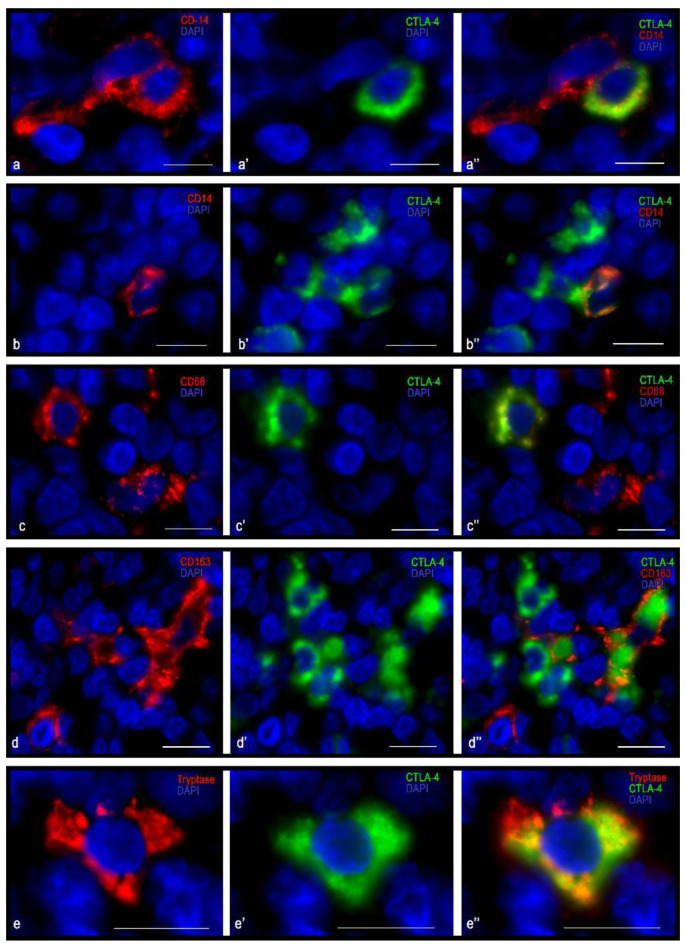
Immunophenotype of CTLA-4(+) tonsillar cells when stained with antibodies of the AA 57-86 clone. (**a**–**a″**,**b**–**b″**) Expression of CD14 in CTLA-4 in cells of the extrafollicular regions of the tonsil. (**c**–**c″**,**d**–**d″**,**e**–**e″**) CTLA-4(+) cells co-expressing CD68, CD163, and tryptase, respectively. Scale bar 10 µm.

**Table 1 cells-10-01027-t001:** Primary antibodies used in this study.

Antibodies/Clone	Host	Source	Dilution
CD1a (EP362R)	rabbit monoclonal Ab	Cell Marque, USA	Ready-to-use
CD3 (EPS1)	mouse monoclonal Ab	AbCam, United Kingdom	1:50
CD3 (E2GV6)	rabbit monoclonal Ab	Ventana/Roche, Germany	Ready-to-use
CD4 (SP35)	rabbit monoclonal Ab	Ventana/Roche, Germany	Ready-to-use
CD8 (ESP57)	rabbit monoclonal Ab	Ventana/Roche, Germany	Ready-to-use
CD14 (EPR3653)	rabbit monoclonal Ab	Cell Marque, USA	Ready-to-use
CD15 (MMA)	mouse monoclonal Ab	Ventana/Roche, Germany	Ready-to-use
CD16 (SP175)	rabbit monoclonal Ab	Cell Marque, USA	Ready-to-use
CD20 (L26)	mouse monoclonal Ab	Ventana/Roche, Germany	Ready-to-use
CD34 (QBEnd/10)	mouse monoclonal Ab	Ventana/Roche, Germany	Ready-to-use
CD38 (SP149)	rabbit monoclonal Ab	Cell Marque, USA	Ready-to-use
CD45 (LCA)	mouse monoclonal Ab	Cell Marque, USA	Ready-to-use
CD57 (NK-1)	mouse monoclonal Ab	Cell Marque, USA	Ready-to-use
CD68 (EKP-1)	mouse monoclonal Ab	Ventana/Roche, Germany	Ready-to-use
CD117 (A4502)	rabbit monoclonal Ab	DAKO/Leica	Ready-to-use
CD138 (B-A38)	mouse monoclonal Ab	Cell Marque, USA	Ready-to-use
CD163 (MRQ-26)	mouse monoclonal Ab	Cell Marque, USA	Ready-to-use
Tryptase (AA1)	mouse monoclonal Ab	AbCam, UK	1:3000
Podoplanin (D2-40)	mouse monoclonal Ab	Cell Marque, USA	Ready-to-use
Vimentin (V9)	mouse monoclonal Ab	Ventana/Roche, Germany	Ready-to-use
Pan Cytokeratin (AE1/AE3/PCK26)	mouse monoclonal Ab	Ventana/Roche, Germany	Ready-to-use
CTLA-4 (UMAB249)	mouse monoclonal Ab	Origene Technologies	1:50
CTLA-4 (AA 57-86)	rabbit polyclonal Ab	antikoerper-online, Germany	1:200
CTLA-4 (SP355)	rabbit monoclonal Ab	AbCam, United Kingdom	1:100
CTLA-4 (CAL49)	rabbit monoclonal Ab	AbCam, United Kingdom	1:300

**Table 2 cells-10-01027-t002:** Secondary antibodies and other reagents.

Antibodies and Other Reagents	Source	Dilution	Label
Goat anti-mouse IgG Ab (#A21236)	Invitrogen, Darmstadt, Germany	1/200	Alexa Fluor 647
Goat anti-rabbit IgG Ab (#A21245)	Invitrogen, Darmstadt, Germany	1/200	Alexa Fluor 647
Goat anti-mouse IgG Ab (#115-165-166)	Jackson ImmunoResearch	1/200	Cy3
Goat anti-rabbit IgG Ab (#A-11034)	Invitrogen Darmstadt, Germany	1/200	Cy3
Goat anti-mouse IgG Ab (#A-11029	Invitrogen Darmstadt, Germany	1/200	Alexa Fluor 488
Goat anti-rabbit IgG Ab (#A-11034)	Invitrogen Darmstadt, Germany	1/200	Alexa Fluor 488
AmpliStain anti-Mouse 1-Step HRP (#AS-M1-HRP)	SDT GmbH, Baesweiler, Germany	ready-to-use	HRP
AmpliStain anti-Rabbit 1-Step HRP (#AS-R1-HRP)	SDT GmbH, Baesweiler, Germany	ready-to-use	HRP
4′,6-diamidino-2-phenylindole (DAPI, #D9542-5MG)	Sigma, Hamburg, Germany	5 µg/ml	w/o
VECTASHIELD Mounting Medium (#H-1000)	Vector Laboratories, Burlingame, CA, USA	ready-to-use	w/o
CC2 solution (# 950-223)	Ventana	ready-to-use	w/o
TSA Plus Fluorescein (#NEL741E001K)	PerkinElmer, Rodgau Germany	1/200	Fluorescein
TSA Plus Cyanine 3 (#NEL744E001KT)	PerkinElmer, Rodgau Germany	1/200	Cyanine 3
TSA Plus Cyanine 5 (#NEL745E001KT)	PerkinElmer, Rodgau Germany	1/200	Cyanine 5

**Table 3 cells-10-01027-t003:** CTLA-4-positive cells in tonsillar compartments.

CTLA4 Clones			UMAB249	AA 57-86	SP355	CAL49
Overall count of CTLA-4-positive cells in the human tonsil			29.4/mm^2^	3.0/mm^2^	10.5/mm^2^	127.4/mm^2^
Comparative distribution of CTLA-4-positive cells in tonsillar compartments						
Reticulated crypt epithelium (lymphoepithelium)			**+++**	**++**	*****	*****
Stratified squamous non-keratinized epithelium			**+++**	**+**	*****	*****
Lymphoid follicle	Mantle zone		**−**	**−**	*****	**+++**
	Germinal center	Dark zone	*****	**−**	**+**	**+++**
		Apical light zone	**−**	*****	**+**	**++**
		Basal light zone	**−**	*****	**+**	**++**
Extrafollicular region			**+++**	**+++**	**+++**	**+++**

Notes: * Rare single cells.

## Data Availability

All data and materials are available upon reasonable request. Address to I.B. (email: buchwalow@pathologie-hh.de) or M.T. (email: mtiemann@hp-hamburg.de) Institute for Hematopathology, Hamburg, Germany.

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
