# Peer review of "Identification of CTLA-4-Positive Cells in the Human Tonsil"

_cells, 2021, doi:10.3390/cells10051027_

Round 1

Reviewer 1 Report

I do think that the paper is of a good value, but before reviewing it, it needs extensive language editing.

Author Response

REV. Nr. 1: Comments and Suggestions for Authors

I do think that the paper is of a good value, but before reviewing it, it needs extensive language editing.  Response: I will load the English language editing by MDPI 

Reviewer 2 Report

The study of Tiemann et al. analyze the distribution pattern of CTLA-4 in the human tonsil performing fluorescent immunolabelling of this protein with antibodies of different clones.

The imaging is very impressive and well done.

Some minor comments:

  • Please provide the negative control image for each different antibody used.
  • No functional studies are performed. For this reason  the role of CTLA-4 in immuno regulation can't be established. In the discussion line 368 and line 379 shoud be rewrited

Author Response

REV. Nr. 2: Comments and Suggestions for Authors

The imaging is very impressive and well done.

Some minor comments:

    Please provide the negative control image for each different antibody used.  Response: According to the Good Laboratory Practice Guidelines, each immunohistochemical experiment includes obligatory negative controls and we have done it, as stated in the Section “2.5. Controls”.  In modern publishing practice it is unusual to back up the textual statement with figures showing that the exclusion of either the primary or the secondary antibody from the immunohistochemical reaction, the substitution of primary antibodies with the corresponding IgG at the same final concentration resulted in a lack of immunostaining.  The TSA step alone did not contribute to any specific immunostaining that might have influenced the analysis.  Many additional figures could only unnecessarily overload the manuscript without bringing additional information.  

No functional studies are performed. For this reason the role of CTLA-4 in immuno regulation can't be established. In the discussion line 368 and line 379 shoud be rewrited should be rewritten.  Response:  Thanks a lot. According to this comment of the Reviewer, the sentence “These findings suggest more extensive effects of CTLA-4 in immune regulation” will be removed from the Discussion.

Reviewer 3 Report

This work identifies the expression of the antigen CTLA-4 in a wide number of cell-lineages using immunohistochemistry and different monoclonal antibodies. It´s a detailed descriptive study which allows some hypothesis about its meaning, but no functional conclusions can be inferred. The authors should be less categorical in sentences like “The expression of CTLA-4 in cells of different origins indicates the involvement of this molecule not only in the well-known classical scheme for the regulation of T lymphocyte activity, but also implies the existence of other CTLA-4 signaling pathways in cancer immunotherapy and suggests more extensive effects of CTLA-4 in immune regulation” at the end of the abstract and another very similar at the end of the Introduction. The findings they describe cannot lead to such conclusions. Verbs like “implies” or “indicates” should be avoided.  

In the same line, the term “check-point” protein should be avoided to describe the CTLA-4 expression in any other cell line different from T-cells. At the moment, the function and signal pathway remain unknown. For the same reason, it is not possible to deduce the existence of various CTLA-4 signaling pathways (Discussion, page 21, line 344-346) and that “…lymphocytes that were prone to contact with other specific cellular elements of the tonsil tissue microenvironment for the development of immune response” in the last sentence of page 15.

Specific comments

  • The sentence “Different immunolabeling patterns with anti-CTLA-4 antibodies of different clones can be explained by generating anti-CTLA-4 antibodies that recognize different epitopes of this protein” is repeated in the Discussion section in lines 358-361 and 369-371. (Please, delete one of them). Three reagents against CTLA-4 are IgG mAb but the clone AA57-86 is the only polyclonal antibody used. Could this technical issue also explain the broad spectrum of cells detected with this last clone?
  • In last sentence of page 9, and Figure 2k, CD34+ cells are classified as “undifferentiated myelopoiesis cells”. Considering that human tonsils may contribute to lymphoid development, the term “myelopoiesis” should be replaced by “hematopoietic”. J Clin Invest. 2012;122(4):1403-1415. https://doi.org/10.1172/JCI46125.
  • Has any objective criterion been used to describe the different intensity of staining described in Table3?
  • Where are located the CTLA-4+ neutrophils specified in the abstract and which antibody identifies them?
  • What do the authors specifically mean when they say “The existence of various CTLA-4 signaling pathways in the specific tissue microenvironment should be taken into account in the diagnosis and targeted therapy of cancer” (Discussion, page 21, lines 344-346? How might this affect diagnosis or target therapy?

Minor comments

  • The letter “A” is missing from CTLA-4 in page 7, line 149 and page18, line 307.
  • In Figure 8c”, the second block of reagents (CTLA-4, CD14, DAPI) should be removed.
  • Please homogenize the term CTLA-4 positive cells (or CTLA-4+ cells) through all the manuscript and legends of Figures.
  • Legends of several figures show a mixture of number and letters, bold and no bold fonts, or lower and capital letters, Please, edit the style.
  • The sentence “The use of Ipilimumab for the melanoma treatment was based on Allison’s concept that Ipilimumab binds to CTLA-4 on the surface of T cells and blocks the inhibitory signal, thereby releasing the effector function of the T-lymphocytes [14]” is repeated in page 1 (lines 44-46) and in the Discussion (page 21, lines 349-351).

Author Response

REV. Nr. 3: Comments and Suggestions for Authors

This work identifies the expression of the antigen CTLA-4 in a wide number of cell-lineages using immunohistochemistry and different monoclonal antibodies. It´s a detailed descriptive study which allows some hypothesis about its meaning, but no functional conclusions can be inferred. The authors should be less categorical in sentences like “The expression of CTLA-4 in cells of different origins indicates the involvement of this molecule not only in the well-known classical scheme for the regulation of T lymphocyte activity, but also implies the existence of other CTLA-4 signaling pathways in cancer immunotherapy and suggests more extensive effects of CTLA-4 in immune regulation” at the end of the abstract and another very similar at the end of the Introduction. The findings they describe cannot lead to such conclusions. Verbs like “implies” or “indicates” should be avoided.  

In the same line, the term “check-point” protein should be avoided to describe the CTLA-4 expression in any other cell line different from T-cells. At the moment, the function and signal pathway remain unknown. For the same reason, it is not possible to deduce the existence of various CTLA-4 signaling pathways (Discussion, page 21, line 344-346) and that “…lymphocytes that were prone to contact with other specific cellular elements of the tonsil tissue microenvironment for the development of immune response” in the last sentence of page 15.

Response: 

  • We avoid the term “check-point” protein and substituted it for CTLA-4 in the Discussion.
  • We exclude from the Abstract and Discussion the sentence “The expression of CTLA-4 in cells of different origins indicates the involvement of this molecule not only in the well-known classical scheme for the regulation of T lymphocyte activity, but also implies the existence of other CTLA-4 signaling pathways in cancer immunotherapy and suggests more extensive effects of CTLA-4 in immune regulation” and changed for “The expression of CTLA-4 in cells of different origins support the alternative view that CTLA-4 is not restricted to the lymphoid cell lineage and can provide broader effects of CTLA-4 on immune regulation.”
  • We exclude from the page 15 the sentence “Thus, CTLA-4 (CAL49 clone) antibody specifically detected lymphocytes that were prone to contact with other specific cellular elements of the tonsil tissue microenvironment for the development of immune response”.

Specific comments

  • of The sentence “Different immunolabeling patterns with anti-CTLA-4 antibodies different clones can be explained by generating anti-CTLA-4 antibodies that recognize different epitopes of this protein” is repeated in the Discussion section in lines 358-361 and 369-371. (Please, delete one of them). Response:  We deleted one of the two sentences “Different immunolabeling patterns with anti-CTLA-4 antibodies of different clones can be explained by generating anti-CTLA-4 antibodies that recognize different epitopes of this protein”.  .  Three reagents against CTLA-4 are IgG mAb but the clone AA57-86 is the only polyclonal antibody used. Could this technical issue also explain the broad spectrum of cells detected with this last clone?  At the moment, we have no immediate answer why the clone AA57-86, the only polyclonal antibody used, detects a broader spectrum of cells.  Possibly, this reflects the fact that polyclonal antibodies are a heterogeneous mix of antibodies, derived from the immune response of multiple B-cells, and each one recognizes a different epitope on the same antigen.
  • In last sentence of page 9, and Figure 2k, CD34+ cells are classified as “undifferentiated myelopoiesis cells”. Considering that human tonsils may contribute to lymphoid development, the term “myelopoiesis” should be replaced by “hematopoietic”. J Clin Invest. 2012;122(4):1403-1415. https://doi.org/10.1172/JCI46125. Response:  We correspondingly replace by “hematopoietic” according to the Reviewer comment.   
  • Has any objective criterion been used to describe the different intensity of staining described in Table3?  Response:  The overall count of CTLA-4-positive cells in the human tonsil is described in the Section “2.4. Quantification of CTLA-4-positive cells in tonsillar compartments”.  The different intensity of staining in tonsillar compartments was reviewed by three pathologists independently (MT, IB and DA) using the x20 objective, and upon reaching the consensus, the scoring was assigned from (+++) to (-).  The consensus scoring (+++) meant the largest number of cells with CTLA4 expression in the field of view with the number of CTLA-4-positive cells more than 50.  In the case of detecting cells from 5 to 20, the scoring with the lowest content of cells was set as (+).  With the number of cells averaged from 20 to 50, the average score was assigned as (++). In the cases of detecting 5 or less cells “Rare single cells" scoring was assigned.  According to the Reviewer’s comment, we have placed this information also within the same sections “2.4”.   
  • Where are located the CTLA-4+ neutrophils specified in the abstract and which  antibody identifies them?   Response:  CTLA-4 + neutrophils were detected using rabbit polyclonal Ab AA 57-86 (Antikoerper-online, Germany) and were located mainly within the microvasculature, in contact with the endothelium.
  • What do the authors specifically mean when they say “The existence of various CTLA-4 signaling pathways in the specific tissue microenvironment should be taken into account in the diagnosis and targeted therapy of cancer” (Discussion, page 21, lines 344-346? How might this affect diagnosis or target therapy?  Response:  We have deleted this unclear sentence.

Minor comments

  • The letter “A” is missing from CTLA-4 in page 7, line 149 and page18, line 307.  Response:  Thanks a lot, we shall make corresponding corrections.
  • In Figure 8c”, the second block of reagents (CTLA-4, CD14, DAPI) should be removed. Response:  Thanks a lot, we shall make corresponding corrections in the Figure 8c’’. 
  • Please homogenize the term CTLA-4 positive cells (or CTLA-4+ cells) through all the manuscript and legends of Figures.  Response:  We shall standardize this term for “CTLA-4(+) cells” over the manuscript and Figure Legends and left the term “CTLA-4 positive cells” only for the manuscript title and subtitles.
  • Legends of several figures show a mixture of number and letters, bold and no bold fonts, or lower and capital letters, Please, edit the style.  Response:  Thanks a lot, we shall make corresponding corrections.
  • The sentence “The use of Ipilimumab for the melanoma treatment was based on Allison’s concept that Ipilimumab binds to CTLA-4 on the surface of T cells and blocks the inhibitory signal, thereby releasing the effector function of the T-lymphocytes [14]” is repeated in page 1 (lines 44-46) and in the Discussion (page 21, lines 349-351).   Response: This sentence will be removed from the Discussion.

Round 2

Reviewer 1 Report

The authors have performed the English editing and in my opinion the paper can be published in its present form.